# UWB Localization Based on Improved Robust Adaptive Cubature Kalman Filter

**DOI:** 10.3390/s23052669

**Published:** 2023-02-28

**Authors:** Jiaqi Dong, Zengzeng Lian, Jingcheng Xu, Zhe Yue

**Affiliations:** School of Surveying and Land Information Engineering, Henan Polytechnic University, Jiaozuo 454003, China

**Keywords:** ultra-wideband, NLOS signal, robust filtering, adaptive filtering

## Abstract

Aiming at the problems of Non-Line-of-Sight (NLOS) observation errors and inaccurate kinematic model in ultra-wideband (UWB) systems, this paper proposed an improved robust adaptive cubature Kalman filter (IRACKF). Robust and adaptive filtering can weaken the influence of observed outliers and kinematic model errors on filtering, respectively. However, their application conditions are different, and improper use may reduce positioning accuracy. Therefore, this paper designed a sliding window recognition scheme based on polynomial fitting, which can process the observation data in real-time to identify error types. Simulation and experimental results indicate that compared to the robust CKF, adaptive CKF, and robust adaptive CKF, the IRACKF algorithm reduces the position error by 38.0%, 45.1%, and 25.3%, respectively. The proposed IRACKF algorithm significantly improves the positioning accuracy and stability of the UWB system.

## 1. Introduction

With the increasing need for indoor navigation and positioning services, wireless radio frequency positioning technologies are developing quickly [1], such as ultra-wideband (UWB) [2], Bluetooth [3], Wi-Fi [4], etc., which have a broader application prospect in society. Among them, UWB technology has received increasing attention for its low power consumption, high transmission rate, and strong penetration [5]. Currently, the ranging accuracy of UWB can reach the decimeter level under Line-of-Sight (LOS) conditions [6]. The filtering algorithm can improve the positioning accuracy of the tag in the case of movement. The Kalman filter (KF), particle filter (PF), extended Kalman filter (EKF), cubature Kalman filter (CKF), and unscented Kalman filter (UKF) have been applied to localization techniques [7,8,9,10]. Particularly, CKF has been shown to outperform EKF, UKF, and PF [11].

When localization is performed in a real indoor environment, obstacles such as the human body, concrete walls, glass windows, metal plates, and wooden doors in the transmission path may block and reflect UWB wireless signals, which introduces Non-Line-of-Sight (NLOS) errors [12]. Moreover, in practical applications, the measurement noise may change dynamically at different times, causing the performance of these filters to degrade and become non-convergent [13]. NLOS identification and mitigation has been a research hotspot in the field of UWB localization, and many methods have been proposed to improve localization accuracy [14]. Yang identified NLOS conditions by using the variance feature of the range information [15], but this method requires multiple measurements for a location. In practice, tags are in constant motion, which cannot satisfy the usage condition of this algorithm. In [16,17], channel features are used to identify NLOS signals. Yang proposed an NLOS identification method based on a feature selection strategy by using an input vector machine (IVM) [16]. Wei proposed a multi-input learning (MIL) neural network model based on the channel impulse response (CIR) and time-frequency diagram of CIR (TFDOCIR) to identify NLOS signals in UWB positioning systems [17]. However, channel characteristics are affected by channel conditions and factors, such as the transmission distance and transmission power. Meanwhile, these methods require a large amount of training data and a high commutating power to analyze the signal channel statistics. Therefore, for real-time online applications in unknown environments, these methods suffer from poor real-time performance.

In most situations, NLOS measurements are assumed to be outliers in the locus filter, so they are directly excluded after identification [18]. However, this approach limits the effectiveness of UWB measurements in dense and complex environments: If too many NLOS measurements are excluded, the geometry of the UWB system localization will be severely damaged, and the localization accuracy will be reduced or even unlocalizable. Meanwhile, in the process of target tracking, the randomness of the target movement results in the imprecision of the constructed model and the unknown statistical properties of the system noise [19]. In this case, the filter may suffer from large state estimation errors and even cause divergence of the filter. Robust filtering and adaptive filtering are often introduced into filtering algorithms to improve the positioning accuracy of navigation [20]. For example, Xu proposed an M-estimation-based robust adaptive multi-model combination navigation algorithm to quickly estimate the statistical properties of the measurement noise. In this algorithm, a model set adaptive adjustment strategy is adopted to make real-time corrections to the model transfer probability matrix, and an M-estimation-based robust Kalman filter is introduced to improve the robustness of filtering [21]. Gao adaptively adjusted and updated the prior information through the equivalent weighting matrix and adaptive factors to resist the interference of system model errors on system state estimation, thus improving the accuracy of state parameter estimation [22]. Zhao proposed a robust adaptive CKF (RACKF) to deal with the problem of an inaccurately known system model and noise statistics by introducing the adaptive factor and robust estimation theory [23]. However, robust filtering and adaptive filtering are two opposite strategies, where the former suppresses the observed information error by the predicted value of the motion model, while the latter suppresses the kinematic projection error by the observed information. Besides, robust filtering and adaptive filtering cannot distinguish the sources of errors. If there are errors in the observed information when adaptive filtering is used or in the predicted information when robust filtering is used, the impact of errors on the positioning accuracy cannot be avoided. Therefore, it is necessary to establish a judgment criterion to evaluate the sources of errors and select an appropriate correction method for different sources of errors.

To identify the error sources and select a suitable strategy to alleviate the impact of errors on positioning accuracy, this paper proposes an improved robust adaptive cubature Kalman filter (IRACKF). The contribution of the paper can be summarized as follows:

The algorithm solves the problem of lack of real-time and applicability of traditional recognition schemes in practical applications. Compared with the traditional recognition scheme, the proposed algorithm processes the observation information sample by sample in real-time, so as to update the state of the propagation path simultaneously. At the same time, in practical application, the sliding window recognition scheme does not need to set the algorithm parameters with the change of the actual application scenario.

The algorithm solves the observation information variance problem by the polynomial fitting strategy, which overcomes the problem that the NLOS signal cannot be identified by the difference in the statistical characteristics of the receiver’s observation information in the dynamic environment.

In the application of the algorithm, we designed a stable and reliable selection mechanism based on a sliding window recognition scheme. The effectiveness of the proposed method and the designed structure is verified by high dynamic scene experiments.

The rest of the paper is organized as follows. Section 2 explains the time difference of arrival (TDOA) observation model, Section 3 describes the CKF model, Section 4 introduces the IRACKF model in detail, Section 5 presents the scenarios and observations of the simulation and real experiments, and finally, Section 6 and Section 7 describe the discussion and conclusions of the paper.

## 2. UWB Distance Difference Model

In the TDOA localization system, line-of-sight propagation refers to a propagation method in which radio waves propagate directly from the tag to the receiver within the distance between the tag antenna and the receiver antenna that can be “seen” by each other. In the case that three receivers have been used for tag positioning, the three hyperbolas will intersect at a point, which is the tag position. NLOS errors will produce a time delay phenomenon, so the TDOA hyperbolas will be shifted at this moment, as shown by the dashed line in Figure 1. Meanwhile, the tag will be located in the area surrounded by dashed and solid lines, which will increase the tag positioning error.

Suppose there are $M\left(M\geq 4\right)$ receivers distributed on a two-dimensional plane. The coordinates of the receivers are $\left(x_{i},y_{i}\right)\;,\;i=1,2,\cdots ,M$. The coordinate of the tag is $\left(x,y\right)$. The true distance between the tag and the *i*-th receiver is:(1)$$r_{i}^{0}=\sqrt{{\left(x_{i}-x\right)}^{2}+{\left(y_{i}-y\right)}^{2}}\;,\;i=1,2,\cdots ,M$$

In the LOS condition, due to the random error, the observation distance from the *i*-th receiver to the tag is:(2)$$r_{i}=r_{i}^{0}+n_{i}\;,\;i=1,2,\cdots ,M$$
where $n_{i}$ is additional independent noise that obeys a zero-mean smooth Gaussian random process with a variance of $\sigma_{\mathrm{LOS}}^{2}$.

In the NLOS condition, there are obstacles between the direct distance from the receiver to the tag. Therefore, the signal cannot reach the tag through straight-line transmission but can only propagate through reflection or diffusion effects. The NLOS observation distance from the *i*-th receiver to the tag is:(3)$$r_{i}=r_{i}^{0}+n_{i}+b_{i}\;,\;i=1,2,\cdots ,M$$
where $b_{i}$ is the NLOS error that follows the normal distribution with a mean of $\mu_{\mathrm{NLOS}}$ and a variance of $\sigma_{\mathrm{NLOS}}^{2}$, and $\mu_{\mathrm{NLOS}}$ > 0, $\sigma_{\mathrm{NLOS}}^{2}\gg \sigma_{\mathrm{LOS}}^{2}$.

In this paper, the TDOA between the two receivers to the tag is multiplied by the speed of light to obtain the distance difference as the observed value:(4)$$r_{i,1}=r_{i}-r_{1}=r_{i,1}^{0}+n_{i,1}+b_{i,1}\;,\;i=2,3,\cdots ,M$$

## 3. Cubature Kalman Filter

Since Kalman introduced the “state space method” into the Gaussian filter and proposed the KF, the filtering methods in the Gaussian framework in a recursive form have emerged successively. For applications in nonlinear systems, the EKF is proposed to transform the conditions of linear systems in the KF into more general nonlinear systems, but the EKF is only applicable to weakly nonlinear systems [24]. For strongly nonlinear systems, the Sigma point KF series of methods approximating the Gaussian probability density through deterministic sampling is often used, and one representative method UKF. The CKF proposed by Arasaratmam, a Canadian scholar in 2009, is based on the Sigma point filter and uses the spherical-radial integration criterion to numerically approximate the Gaussian integral and perform recursive state estimation. Compared with UKF, CKF has a stricter mathematical derivation and higher filtering precision.

Considering nonlinear discrete additive noise in the conventional case, the modeling of the positioning system is:(5)$$\left\{\begin{array}{l} x_{k+1}=f\left(x_{k}\right)+w_{k}\\ y_{k+1}=h\left(x_{k+1}\right)+v_{k+1} \end{array}\right.$$
where f and *h* are the nonlinear state transfer function and the measurement function, respectively. $x_{k+1}\in R^{n_{x}}$ and $y_{k+1}\in R^{n_{y}}$ are the system state vector and the measurement vector, respectively. $w_{k}$ and $v_{k+1}$ are the system process noise and the measurement noise, respectively, and they are independent of each other. The corresponding covariance matrices are $Q_{k}$ and $R_{k}$, respectively. The initial state of the system is $x_{0}∼N\left(0,P_{0}\right)$, which is uncorrelated with $w_{k}$ and $v_{k+1}$. $P_{0}$ is the initial covariance matrix.

### 3.1. Prediction Update

For the prediction update, the posterior density function of the system at moment k is $p\left(x_{k}\left|y_{k}\right.\right)=N\left({\hat{x}}_{k\left|k\right.},P_{k\left|k\right.}\right)$. The algorithm flow is as follows.

(1)Decompose the prediction error covariance array $P_{k\left|k\right.}$ as:

(6)$$P_{k\left|k\right.}=S_{k\left|k\right.}{\left(S_{k\left|k\right.}\right)}^{T}$$
where S is the square root coefficient of the covariance matrix.

(2)Calculate the cubature points as:



(7)
$$X_{i,k|k}=S_{k|k}\xi_{i}+{\hat{x}}_{k|k}\;\;\;\;i=1,2,\cdots ,m$$



(3)Calculate the cubature points propagated through the nonlinear state transfer function as:

(8)$$X_{i,k+1|k}^{*}=f\left(X_{i,k+1|k}\right)\;\;\;\;i=1,2,\cdots ,m$$
where $f\left(\cdot \right)$ is the nonlinear state transition function. $X_{i,k|k}$ and $X_{i,k+1|k}^{*}$ are cubature points. m is the number of cubature points, which is twice the dimension of the state vector n according to the third-order cubature principle, i.e., m=2n. $\xi_{i}=\sqrt{m/2}{\left[1\right]}_{i}$ is the basic cubature point set, and ${\left[1\right]}_{i}$ is the *i*-th column of $\left[1\right]$. The point set $\left[1\right]$ can be expressed as:(9)$$[1]=\left\{\left(\begin{array}{c} 1\\ 0\\ \vdots \\ 0 \end{array}\right),\left(\begin{array}{c} 0\\ 1\\ \vdots \\ 0 \end{array}\right),\cdots ,\left(\begin{array}{c} 0\\ 0\\ \vdots \\ 1 \end{array}\right),\left(\begin{array}{c} -1\\ 0\\ \vdots \\ 0 \end{array}\right),\left(\begin{array}{c} 0\\ -1\\ \vdots \\ 0 \end{array}\right),\cdots ,\left(\begin{array}{c} 0\\ 0\\ \vdots \\ -1 \end{array}\right)\right\}$$

(4)Calculate the state prediction values and the prediction covariance matrix as:


(10)
$${\hat{x}}_{k+1|k}=\frac{1}{m}\sum_{i=1}^{m}X_{i,k+1|k}^{*}$$



(11)
$$P_{k+1|k}=\frac{1}{2n}\sum_{i=1}^{2n}X_{i,k+1|k}^{*}{\left(X_{i,k+1|k}^{*}\right)}^{T}-{\hat{x}}_{k+1|k}{\hat{x}}_{k+1|k}^{T}+Q_{k}$$


### 3.2. Measurements Update

(1)Decompose the prediction error covariance matrix $P_{k+1\left|k\right.}$ as:



(12)
$$P_{k+1|k}=S_{k+1|k}{\left(S_{k+1|k}\right)}^{T}$$



(2)Calculate the cubature points as:



(13)
$$X_{i,k+1|k}=S_{k+1|k}\xi_{i}+{\hat{x}}_{k+1|k}\;\;\;\;i=1,2,\cdots ,m$$



(3)Calculate the cubature points propagated through the nonlinear measurement function as:

(14)$$Y_{i,k+1|k}=h\left(X_{i,k+1|k}\right)\;\;\;\;i=1,2,\cdots ,m$$
where $h\left(\cdot \right)$ is the observation function.

(4)Calculate the predicted value of the measurement as:



(15)
$${\hat{y}}_{k+1|k}=\frac{1}{m}\sum_{i=1}^{m}Y_{i,k+1|k}$$



(5)Calculate the innovation and innovation covariance matrix, respectively, as:

(16)$$e_{k+1}=y_{k+1}-{\hat{y}}_{k+1|k}$$(17)$$P_{yy,k+1|k}=\frac{1}{2n}\sum_{i=1}^{2n}Y_{i,k+1|k}Y_{i,k+1|k}^{T}-{\hat{y}}_{k+1|k}{\hat{y}}_{k+1|k}^{T}+R_{k+1}$$
where $y_{k+1}$ is the measured value of the system at the time of k+1.

(6)Calculate the mutual covariance matrix as:



(18)
$$P_{xy,k+1|k}=\frac{1}{2n}\sum_{i=1}^{2n}X_{i,k+1|k}Y_{i,k+1|k}^{T}-{\hat{x}}_{k+1|k}{\hat{y}}_{k+1|k}^{T}$$



(7)Calculate the filter gain $K_{k+1}$, the state estimate ${\hat{x}}_{k+1|k+1}$, and the estimation error covariance matrix $P_{k+1|k+1}$ at the time of k+1, respectively, as:



(19)
$$K_{k+1}=P_{xy,k+1|k}P_{yy,k+1|k}^{-1}$$


(20)
$${\hat{x}}_{k+1|k+1}={\hat{x}}_{k+1|k}+K_{k+1}\left(y_{k+1}-{\hat{y}}_{k+1|k}\right)$$


(21)
$$P_{k+1|k+1}=P_{k+1|k}-K_{k+1}P_{yy,k+1|k}K_{k}^{T}$$



### 3.3. Error Analysis

It can be seen from Equation (20) that the state estimate ${\hat{x}}_{k+1|k+1}$ consists of the priori state prediction ${\hat{x}}_{k+1|k}$ and the posteriori measurement of the innovation feedback $K_{k+1}$ and $e_{k+1}$. When the target motion model at moment k does not match the established model or the target state changes abruptly, and when the sensor observation fails or is disturbed by uncontrollable factors, there is a large deviation between the observed value $y_{k+1}$ and the measured predicted value ${\hat{y}}_{k+1|k}$, which can be obtained by calculating the state predictions at moment k+1 through the nonlinear measurement function. Then, this deviation is reflected in the larger innovation $e_{k+1}$ at moment k+1.

## 4. Improved Robust Adaptive Cubature Kalman Filter

When the NLOS signal is identified, the sensor observation data has low reliability, and the prediction information can be used to correct the observation data through the robust filter. However, when the NLOS propagation of the signal does not occur, the observation data has high reliability, so it can be used to improve the prediction information through the adaptive filter. To identify the error sources and select a suitable strategy to alleviate the impact of errors on positioning accuracy, this paper proposes the IRACKF algorithm, and the first step of the algorithm is to identify the presence of NLOS signals.

### 4.1. The Sliding Window Recognition Scheme Based on Polynomial Fitting

In practical applications, the time and frequency of NLOS signal appearance and disappearance are unknown. Thus, it is necessary to check the LOS/NLOS state of the propagation path at this moment for each observation. In this paper, the sliding window method is used to record m observations before each observation and calculate the variance of these m observations. Based on the difference in the variance characteristics of UWB signals in LOS and NLOS conditions, the LOS/NLOS state of the propagation path of the observed data at this moment can be determined. However, because the tags are in momentary motion and the true distance from each receiver to the tag is not constant, the variance of the observations could not be derived through direct calculation. Therefore, considering that the tag is in the state of momentary motion, this paper proposes a sliding window scheme based on polynomial fitting to identify NLOS signals, and the flow chart of this scheme is shown in Figure 2.

The traditional identification scheme records m pieces of observation data first and then determines whether the propagation path is in the LOS or NLOS state at the next time by solving the variance of these data, as shown in Figure 3. In this figure, the slash-filled part of $\left[0,T\right]$ indicates that the recorded data are used for state checking. After the LOS/NLOS state is determined at moment T, the state will not change until the next state update at N+T. However, T and N in this method are only applicable to specific scenarios, and different values of T and N need to be selected as the scenario changes. Moreover, if the LOS/NLOS state of the propagation path changes while waiting for the state update, it is impossible to replace the appropriate filter, which results in a large error in the estimate.

In this paper, a sliding window identification scheme with real-time sample-by-sample processing and corresponding identification is designed, as shown in Figure 4. The variance of the m pieces of data before each observation is solved to determine the LOS/NLOS state of the propagation path at this moment. The sliding window recognition scheme can respond quickly to changes in the scene, thus achieving real-time recognition when the LOS/NLOS state changes and overcoming the drawbacks caused by the periodic interval checking of the traditional recognition scheme.

However, the true distance from the receiver to the tag changes from time to time as the tag moves. To solve the variance of the observed data in this case, a polynomial fitting scheme is proposed in this paper. This scheme is first performed on the *m* pieces of data before each observation to find the motion law of the observation, after which the fitted value, i.e., the pseudo-true value of the observation, is calculated, and the variance of the observation is solved finally. The steps of the polynomial fitting scheme are as follows.

Step 1:The m pieces of historical observations up to moment *k* are fitted to the h-order. The curve function can be denoted as $F\left(x\right)=\sum_{i=0}^{h}a\left(i\right)*x^{i}$, where a is the coefficient vector of the polynomial fitting function.Step 2:Substitute the abscissas of the m pieces of data into the fitting function to obtain the fitted values corresponding to the data.Step 3:The pseudo-error of the observed value is obtained by subtracting the fitted value obtained from the fitting function and the observed value. Then, the variance of the pseudo-error is calculated and compared with the threshold value to determine the LOS/NLOS state of the propagation path at moment k. The threshold value is the average of the variance calculated from multiple observations of UWB receivers and the tag under LOS conditions.

### 4.2. Robust Cubature Kalman Filter

In the localization process of the UWB system, the propagation path in the NLOS state will bring large errors to the observation data. If the data are substituted into the filtering algorithm, which will seriously reduce the accuracy and reliability of the estimated value. In the CKF, the error of the observations only affects the measurement process of this update. Thus, as for the effect of NLOS errors on the positioning accuracy, only the measurement update part is adjusted by the robust filter, that is, only the autocorrelation covariance matrix of the observations in Equation (17) is adjusted.
(22)$$P_{yy,k+1|k}=\frac{1}{2n}\sum_{i=1}^{2n}Y_{i,k+1|k}Y_{i,k+1|k}^{T}-{\hat{y}}_{k+1|k}{\hat{y}}_{k+1|k}^{T}+{\bar{R}}_{k+1}$$
where, ${\bar{R}}_{k+1}=\alpha R_{k+1}$, and α is the robust factor.

The commonly used methods for calculating the resistance factor include the Huber method, Andrew method, IGGIII method, and Tukey method [25]. For the case that the number of state parameters is larger than the number of observations in the UWB localization model of this paper, the statistics are obtained by using a single prediction residual value. Meanwhile, the expression of the equivalent weight matrix is established using the Huber method to ensure that the diagonal of the equivalence power array is not zero. The statistics obtained based on the prediction residuals are as follows:(23)$$\Delta {\bar{v}}_{i,k}=\frac{{\bar{v}}_{i,k}}{\sigma_{i,k}}$$
where, ${\bar{v}}_{i,k}={\left(y_{k+1}-{\hat{Y}}_{\left.k+1\right|k}\right)}_{i}$ is the component corresponding to the predicted residual vector. $\sigma_{i,k}={\left(P_{yy,\left.k+1\right|k}\right)}_{ii}$ is the diagonal element of the autocorrelation covariance matrix of the observations before correction. The mean $D_{i}$ of $\Delta {\bar{v}}_{i,k}$ is calculated by conducting several experiments in the LOS environment. Then, the robust factor is obtained as:(24)$$\alpha_{i}=\left\{\begin{array}{c} 1,\\ D_{i}/\Delta {\bar{v}}_{i,k}, \end{array}\right.\begin{array}{c} \mathrm{LOS}\;state\\ \mathrm{NLOS}\;state \end{array}$$

Let $\alpha =diag[\alpha_{1},\alpha_{2},\cdots ,\alpha_{L}]$, where L is the number of measured values.

### 4.3. Adaptive Cubature Kalman Filter

The robust cubature Kalman filter (RCKF) is developed based on the CKF by adding the robust factor to adjust the autocorrelation covariance matrix of the observations. When the NLOS state is not detected in the observations, the RCKF automatically degrades to the CKF. However, events occur when the target motion model does not match the established model or when the target state changes abruptly, leading to an inaccurate covariance matrix of the system noise. To improve the stability and accuracy of the filtering in such cases, an adaptive cubature Kalman filter (ACKF) is proposed by combining the Sage–Husa adaptive filter with CKF.

When the receiver in the tag propagation path is in the LOS state, the reliability of the observations is high. To suppress the kinematic projection error by using the observation information, this paper estimates the system noise covariance matrix in ACKF in real time.
(25)$$\chi_{k+1|k}^{*}\chi_{k+1|k}^{*T}=\frac{1}{2n}\sum_{i=1}^{2n}X_{i,k+1|k}^{*}{\left(X_{i,k+1|k}^{*}\right)}^{T}-{\hat{x}}_{k+1|k}{\hat{x}}_{k+1|k}^{T}$$
(26)$${\hat{Q}}_{k+1}=\left(1-d_{k}\right){\hat{Q}}_{k}+d_{k}\left(K_{k+1}e_{k+1}e_{k+1}^{T}K_{k+1}^{T}+P_{k+1}-\left.\chi_{k+1|k}^{*}\chi_{k+1|k}^{*T}\right)\right.$$
where $d_{k}=\left(1-b\right)/\left(1-b^{k}\right)$. b is the forgetting factor with a value of 0.99.

### 4.4. Improved Robust Adaptive Cubature Kalman Filter

The pseudocode of the proposed IRACKF is shown in Algorithm 1.

**Algorithm 1:** IRACKF.**Input**: Initial state $\left[x_{0},y_{0},v_{x0},v_{y0}\right]$, sampling number *T* observation dimension *n* prediction noise covariance matrix ***Q***, observation noise covariance matrix ***R***, error covariance matrix of the priori state ***P***, variance threshold *Var*_*LOS*, the average value of
$\Delta \bar{v}$ in LOS environment ***D***.**Output**: Estimate position and velocity
$\left[X_{1}\;X_{2}\;\ldots \;X_{T}\right]$1: $X_{1}$$\text{←}\left[x_{0},y_{0},v_{x0},v_{y0}\right]$2: **for**
*t* ← 2 to *T*
**do**
3: $Y_{t}$ ← Read UWB observations
4: // Calculate the variance of errors in UWB observations
5:  **Var** ← get_var
$\left(Y_{t}\right)$
6: // Prediction update
7: // Calculate the state prediction values and the prediction covariance matrix 8: $\left(x_{t},p_{t}\right)$ ← get_pre($X_{t-1}$$,P_{t-1}$)
9: // Calculate the robust factor
10: // Calculate the innovation and innovation covariance matrix
11: $(e_{t}$$,py_{t}$$)\;\text{←}\;\mathrm{get}\text{\_}\mathrm{py}(x_{t}$$,p_{t}$$,Y_{t}$)
12: α← identity matrixfor
13: **for**
*j* ← 1 to *n*
**do**
14: if **Var**$\left(j\right)$ > *Var_LOS*
15: $\Delta {\bar{v}}_{j,t}$$\text{←}\frac{e_{j}}{py_{jj}}$
16: $\alpha \left(j,j\right)$$\text{←}\frac{D_{j}}{\Delta {\bar{v}}_{j,t}}$
17: **end if**
18: *j* ← *j* + 1
19: **end for**
20: // Measurements update
21: **if** det$\left(\alpha \right)$<>1
22: $R_{t-1}$
$\text{←}\alpha *R_{t-1}$
23: $\left(X_{t},P_{t}\right)$←Mu_RCKF$\left(x_{t},p_{t},Q_{t-1},R_{t-1},Y_{t}\right)$
24: **else**
25: $\left(X_{t},P_{t},Q_{t}\right)$←Mu_ACKF$\left(x_{t},p_{t},Q_{t-1},R_{t-1},Y_{t},t\right)$
26: **end if**
27: *t* ← *t* + 1
28: **end for**


The flow chart of the IRACKF is shown in Figure 5.

## 5. Experimental Analyses

### 5.1. Simulation Experiments

In this section, MATLAB 2019b is used to simulate and analyze the IRACKF algorithm, and all results are obtained on the same computer equipped with a 3.4 GHz CPU and 8 GB RAM. The simulation scenario and parameters are set as follows: Eight receivers are deployed in the range of 20 × 20 m with receiver coordinates of (0,0), (0,10), (0,20), (10,20), (20,20), (20,10), (20,0) and (10,0), respectively.

#### 5.1.1. Experimental Evaluation of CKF

To prove that CKF performs better in localization navigation, this section compares CKF with PF and unscented particle filter (UPF) in a control experiment. In the simulation scenario in the LOS environment, the tag is moved in uniformly and linearly from [0,0] to [20,20], and eight receivers localize the tag to obtain the distance difference information from the receivers to the tag. With the same observation information and motion model, the estimates of tag position and velocity are solved for each moment, and the results are shown in Figure 6. It can be seen that compared to PF and UPF, the CKF reduces the position error by 49.6% and 21.5% and the velocity error by 65.0% and 54.6%, respectively. Therefore, the CKF achieves better positioning accuracy and stability than PF and UPF.

#### 5.1.2. Adaptive Filtering Effectiveness Analysis

When the target motion model does not match the established model or the system noise covariance matrix is inaccurate, the prediction information and the prediction error covariance array $P_{k\left|k\right.}$ is less reliable. However, the CKF algorithm is not sensitive to the prediction information to know the error change of the observation data and update the prediction error covariance array $P_{k\left|k\right.}$ in real time. Thus, in this paper, the Sage–Husa adaptive filter is adopted to adjust the prediction error covariance array $P_{k\left|k\right.}$ by using more reliable observation data. To verify the effect of ACKF on the localization accuracy, the following experiment is conducted: The UWB tag still travels following the above path, and compared with the experimental data in Section 5.1.1, the error probability density of the observation information remains unchanged, and the covariance matrix of the prediction information changes randomly. The experimental results are shown in Figure 7.

It can be seen from Figure 7 that the position error of ACKF is smaller than that of CKF. The average position errors of ACKF and CKF are 6.5 and 9.3 cm, respectively. Besides, the localization accuracy is improved by 30.1%. Therefore, the Sage–Husa adaptive filtering can improve the localization accuracy of the CKF when the prediction information error is large.

#### 5.1.3. Robust Filter Effectiveness Analysis

In practical applications of UWB systems, the receiver-to-tag distance difference is often contaminated by NLOS error. To reduce the impact of NLOS errors on localization accuracy, this paper introduces the robust factor into the CKF and uses the prediction information to correct the observation data when the target motion model matches the established model, and the target state does not change abruptly. To analyze the weakening effect of the RCKF on the NLOS error, the following experiment is conducted: The UWB travel path is unchanged, and compared with the experimental data in Section 5.1.1, the covariance matrix of the prediction information is unchanged, and the NLOS error is added to the observation data. The experimental result is shown in Figure 8.

As can be seen from Figure 8, the NLOS error increases the position error of the CKF, and the introduction of robust filtering alleviates the effect of NLOS error on the localization accuracy. After calculation, the average values of the position errors of CKF and RCKF are 16.9 and 8.1 cm, respectively, and the positioning accuracy is improved by 52.1%.

#### 5.1.4. IRACKF Effectiveness Analysis

The above analysis indicates that robust filtering and adaptive filtering can alleviate the impact of observation and prediction errors on positioning accuracy. However, the occurrence of events, such as NLOS signals or sudden changes in target status during actual UWB localization, cannot be effectively identified. Therefore, to identify the error sources, a sliding window scheme based on polynomial fitting is proposed in this paper.

To verify the effectiveness of the polynomial fitting method for solving the variance of the observations, this paper first extracts m observations before moment k, after which the variance of the observations is calculated and compared with the threshold value to determine whether the propagation path is in the NLOS state at moment k.

To demonstrate the effect of the polynomial fitting method, a set of NLOS data is selected and fitted in this paper, and the experimental results are presented in Figure 9.

It can be found from Figure 9 that the observed value in the NLOS state is much larger than the true value. Meanwhile, the movement of the tag makes the distance difference from the receiver to the tag vary regularly. Therefore, to find the variance of the observation data, this paper obtains the pseudo-truth value of the observation data by polynomial fitting, i.e., the difference between the pseudo-truth value and the corresponding truth value is an almost constant amount. Then, the error between the observed value and the pseudo-true value is calculated, and the variance of the calculated error is the variance of the observed value. Finally, the sliding window scheme based on polynomial fitting is performed for each observation in the filtering to obtain the LOS/NLOS state of the propagation path at this moment.

To verify the effectiveness of the IRACKF, the experimental data in Section 5.1.2 are taken as the basis, and the NLOS error is added to the observation data. Besides, the receiver is set to position five times per second. The filtering algorithm takes the last observation data of the receiver per second as the observation data in the filtering. The NLOS error is added to $r_{3,1}$ and $r_{6,1}$ in the simulation of the observation data, and the observation error is shown in Figure 10.

Based on the experimental data in Section 5.1.2, NLOS errors are added to the ob-served data. To verify the effectiveness of IRACKF, the signal tracking experiment is compared based on the generated data with five algorithm combinations: CKF, RCKF, ACKF, RACKF, and IRACKF. The RACKF and IRACKF algorithms differ in that the IRACKF algorithm has a sliding window identification scheme that can identify NLOS signals in real-time.

The observation data in the simulation contains NLOS errors, and the covariance matrix of the prediction information in the filtering cannot accurately describe the noise when the prediction information of the system is updated. To alleviate the impact of NLOS errors on the positioning accuracy and correct the prediction covariance matrix by the observation data when there is no NLOS error interference, the data are processed by the IRACKF algorithm. Then, to reflect the advantages of the IRACKF, CKF, RCKF, and ACKF are respectively added to the comparison algorithm in this paper, and the experimental results are shown in Figure 11.

It can be seen from Figure 11 that at the moments when NLOS signals are present, the sliding window scheme based on polynomial fitting can identify all these moments and corrects the covariance matrix of the observation data by robust filtering. The position errors of the RCKF, RACKF, and IRACKF are much smaller than those of the CKF and ACKF at these moments. IRACKF, RACKF, and ACKF add Sage–Husa adaptive filtering based on CKF and correct the prediction information by the observation data with higher accuracy, and the localization accuracy is better than that of CKF and RCKF algorithms. RACKF can deal with both abnormal measurement noise and inaccurate motion model problems. Therefore, its tracking stability is better than that of ACKF and RCKF, which is relatively stable during the whole tracking process. However, due to the lack of an error identification mechanism and the different conditions for the application of robust and adaptive filtering, there is still room for further improvement in the positioning accuracy of the RACKF algorithm. After calculation, the average position errors of CKF, RCKF, ACKF, RACKF, and IRACKF are 13.5, 10.0, 11.3, 8.3, and 6.2 cm, respectively. Compared with CKF, RCKF, ACKF and RACKF, IRACKF reduces the position error by 52.6%, 38.0%, 45.1%, and 25.3%, respectively.

### 5.2. Field Experiments

To further verify the effectiveness of the proposed algorithm, the DWM1000 UWB localization module was selected for field experiments. The experiment site is located in the lobby on the first floor of the experiment building, surrounded by classrooms and an aisle. In this indoor positioning condition, the thick transparent glass above the hall attenuates the GNSS signal. In this experiment, the experimental range was determined to be 9 m*9 m. To obtain the tag coordinates, seven modules were selected, six of which were used as receivers to locate the tag, and the receivers could locate the tag five times per second. The last measurement per second was selected as the observation data. In the experiment, the true positions of the receiver and tag are obtained by total station observation. The tag moves forward along the predetermined trajectory, and the experimental scene and the moving trajectory of the tag are presented in Figure 12.

During the experiment, the experimenters moved irregularly in the field while the receiver positioned the tag. Therefore, the UWB observation information is affected by NLOS errors. Meanwhile, there is an error between the moving trajectory of the tag and the intended trajectory, which makes the covariance matrix of the predicted information unknown. To obtain high-precision positioning results, this paper sets the experimental data by IRACKF, and the resultant path and position errors are illustrated in Figure 13 and Figure 14, respectively.

From Figure 14, it can be seen that when the observation information is contaminated by NLOS errors, the position error of the tag increases sharply, while the proposed algorithm is less affected by NLOS errors owing to the addition of robust filtering. Adaptive filtering also reduces the impact of tag movement path errors on localization accuracy. The average position errors of CKF and IRACKF are 17.8 and 6.2 cm, respectively. Compared with that of CKF, the localization accuracy of the proposed algorithm is improved by 65.2%.

## 6. Discussion

In the process of target tracking and localization, the filtering algorithm is often susceptible to NLOS errors and inaccurate system noise covariance matrices. To identify the error sources and select a suitable filter for error handling, this paper proposes the IRACKF. In simulations and field experiments, the proposed algorithm can identify the LOS/NLOS states of the propagation path and select different methods to process NLOS errors and inaccurate covariance matrices, which improves the positioning accuracy of UWB systems in complex situations.

In a comparative test between the CKF and PF and UPF, the CKF can achieve better estimation accuracy regardless of position and velocity. Compared with PF and UPF, the CKF reduces the position error by 49.6% and 21.5% and the velocity error by 65.0% and 54.6%. In this paper, a sliding window scheme based on polynomial fitting, robust filtering, and adaptive filtering is added to the CKF. When both NLOS error and inaccurate system noise covariance matrices are present, compared with CKF, RCKF, ACKF, and RACKF, the proposed algorithm improves the localization accuracy by 52.6%, 38.0%, 45.1%, and 25.3%, respectively. Therefore, IRACKF can improve the localization accuracy of UWB systems. The results of field experiments indicate that the target trajectory solved by IRACKF is closer to the real trajectory than that of CKF, and the position error is reduced by 69.4%. Thus, the effectiveness of the IRACKF is further verified.

For NLOS errors and inaccurate system noise covariance matrices, robust filtering and adaptive filtering have been proposed. However, due to the indoor complex environment when the UWB system is actually used, the appearance of NLOS signals and the trajectory of tag movement are random and contingent, while different error types need to be processed by different schemes to reduce the impact of errors on positioning accuracy. The current filtering algorithms fail to analyze error sources when processing errors. Therefore, before error processing, this paper first analyzes the measured values of the receiver through a sliding window scheme based on polynomial fitting to identify the LOS/NLOS state of the signal propagation path. In this way, the error source is determined, and this scheme also overcomes the deficiency that the variance characteristics of the UWB signal cannot be applied in dynamic environments. Afterward, by using robust filtering and adaptive filtering to reduce the effect of NLOS error and inaccurate system noise covariance matrices on positioning accuracy, this paper establishes a selection mechanism to weaken the error effect for different error sources and improves the positioning accuracy of the UWB system.

However, there are still defects in the study. For example, in the simulation of NLOS signals, the probability density of NLOS error is set to Gaussian distribution with a mean value greater than 0. However, in practical applications, the NLOS error may obey exponential, uniform, Gaussian, or δ-distribution under different channel environments. Even if the NLOS error obeys Gaussian distribution, different application scenarios may result in different mean and variance values of the NLOS error. However, the variances are all significantly different from the measured values in the LOS state. Therefore, this paper conducts experiments on the DWM1000 UWB receiver to derive the probability density of its measured values in the NLOS state and use it to represent the distribution characteristics of the NLOS error.

The localization accuracy of UWB systems can meet the accuracy requirements in most situations. However, the combination of IMU, vision, or other sensors for navigation can significantly improve localization accuracy and stability. Therefore, to achieve better positioning performance, further research needs to be conducted to combine UWB with other sensors through filtering algorithms.

## 7. Conclusions

To meet the requirement of high accuracy of tags in moving situations, CKF is applied in localization techniques, but the problems of NLOS errors and inaccurate prediction information seriously degrade the localization accuracy of filtering. To improve the localization accuracy of filtering, this paper proposes the IRACKF algorithm. First, to accurately identify the NLOS signal, the observation information is processed in real-time sample-by-sample by the sliding window scheme based on polynomial fitting. Then, the observation data contaminated by NLOS errors are adjusted by the robust filtering to the autocorrelation covariance matrix to alleviate the impact of NLOS errors on the localization accuracy. Finally, to improve filtering accuracy and stability, for the observation data in the LOS state, the adaptive filtering is used to estimate and correct the statistical characteristics of the system noise in real-time. The results of simulation and field experiments indicate that:

(1)By using robust filtering and adaptive filtering, the position errors are reduced by 52.07% and 30.11% by correcting the observation data containing NLOS errors and inaccurate covariance matrices, respectively. Therefore, robust filtering and adaptive filtering can improve the localization accuracy of CKF.(2)By analyzing the variance characteristics of the observed information, the sliding window scheme based on polynomial fitting can identify the LOS/NLOS state of the propagation path. Based on this, the data are solved by the IARCKF, and the position error is reduced by 52.6%, 38.0%, 45.1%, and 25.3%, compared to CKF, RCKF, ACKF, and RACKF, respectively. Therefore, the IARCKF can identify the error sources and utilizes suitable filtering algorithms to significantly improve the tag localization accuracy.(3)Field experiments further verify the effectiveness of the proposed algorithm. Compared with CKF, the tag movement path of IRACKF is closer to the real path, and the position error is reduced by 65.2%. Thus, the IRACKF algorithm can obtain higher localization accuracy and stability in practical applications.

## Figures and Tables

**Figure 1 sensors-23-02669-f001:**
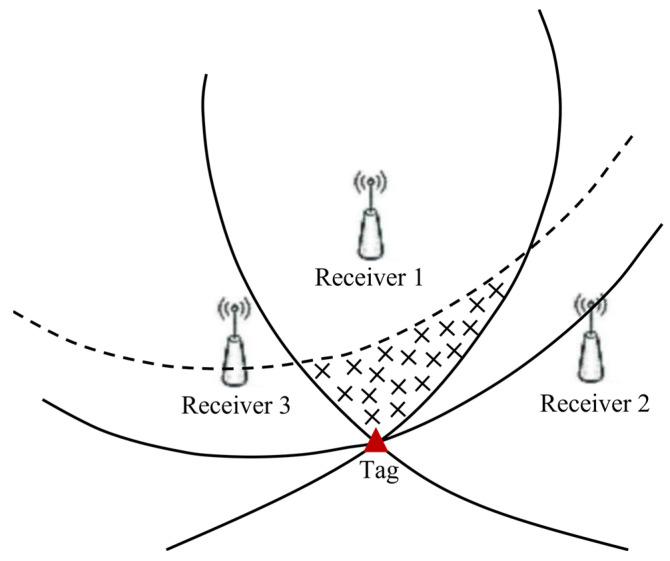
The effect of NLOS error on positioning accuracy.

**Figure 2 sensors-23-02669-f002:**
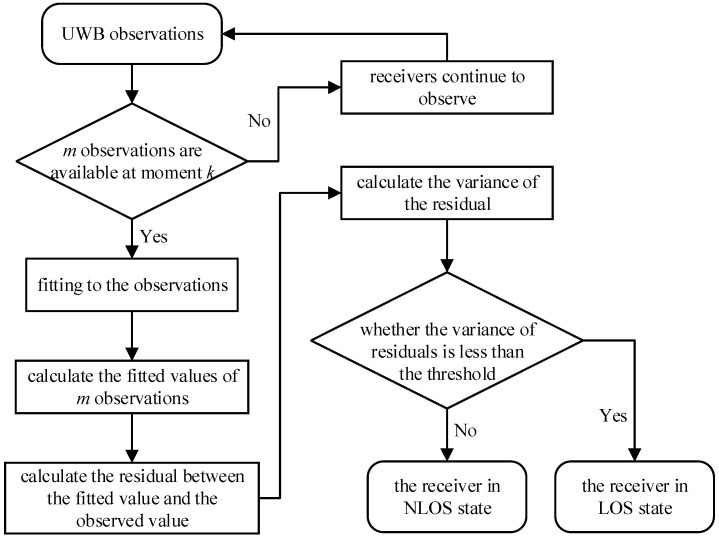
The sliding window recognition scheme based on polynomial fitting.

**Figure 3 sensors-23-02669-f003:**
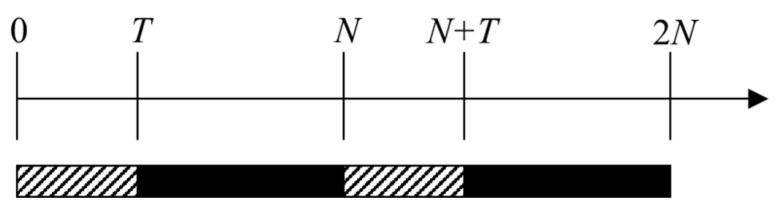
The traditional identification scheme.

**Figure 4 sensors-23-02669-f004:**
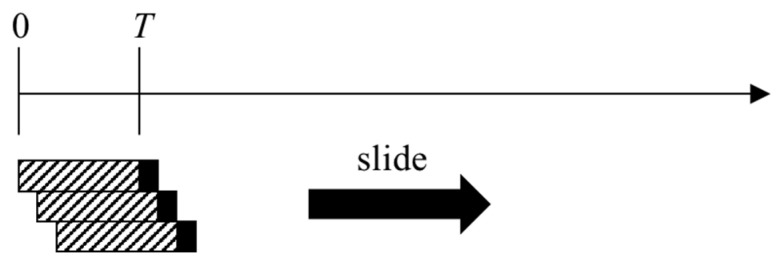
The sliding window identification scheme.

**Figure 5 sensors-23-02669-f005:**
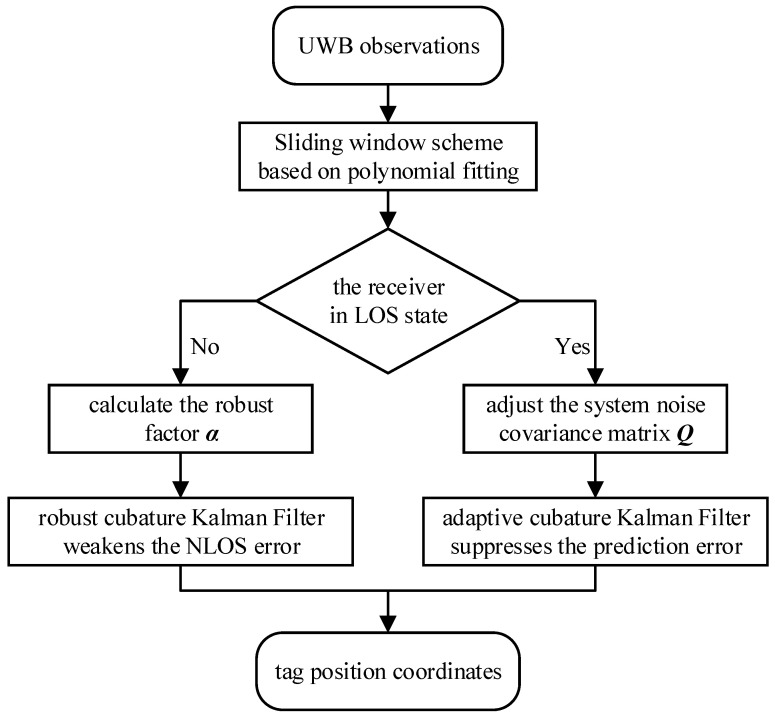
The flow chart of the IRACKF.

**Figure 6 sensors-23-02669-f006:**
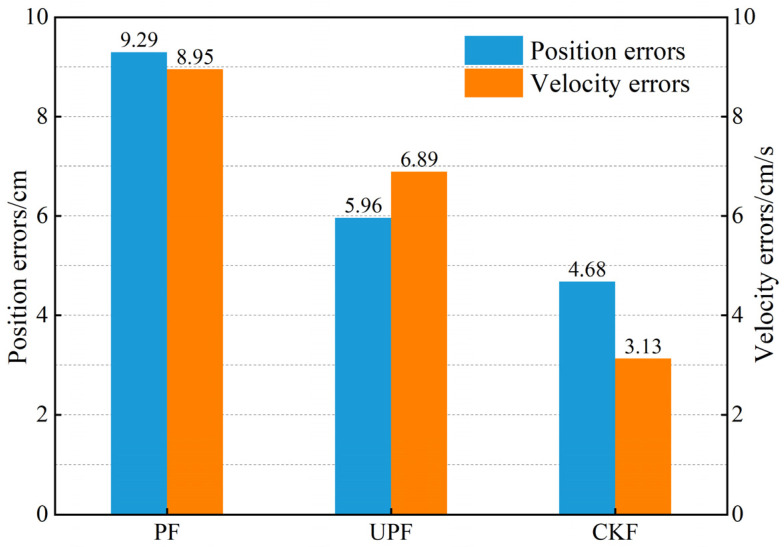
The results of the filtering experiment.

**Figure 7 sensors-23-02669-f007:**
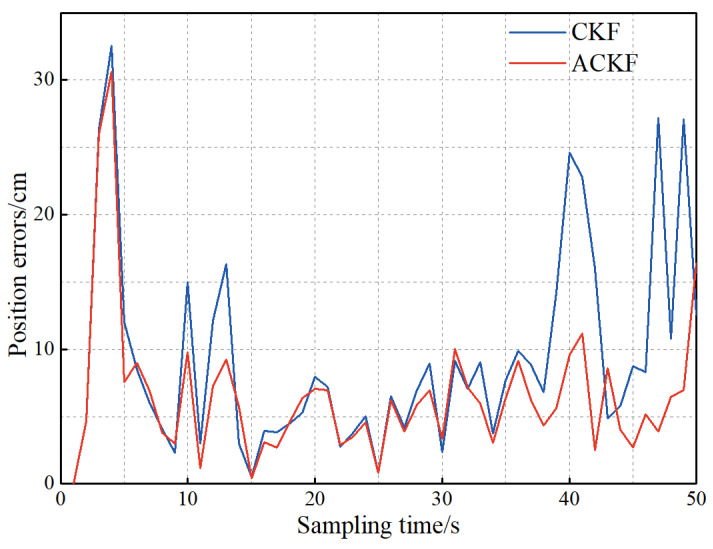
The influence of adaptive filtering on positioning accuracy.

**Figure 8 sensors-23-02669-f008:**
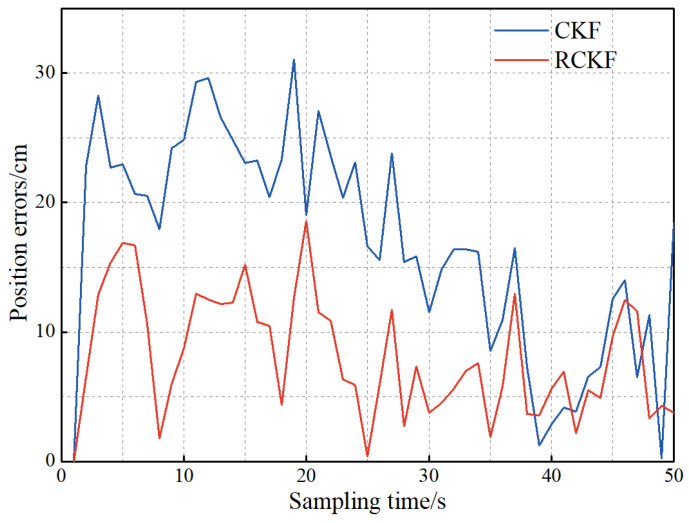
The influence of robust filtering on positioning accuracy.

**Figure 9 sensors-23-02669-f009:**
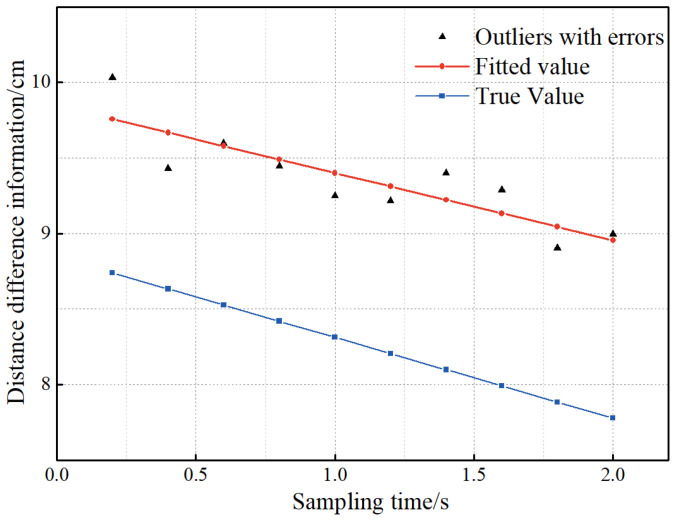
The fitting effect of the polynomial fitting method.

**Figure 10 sensors-23-02669-f010:**
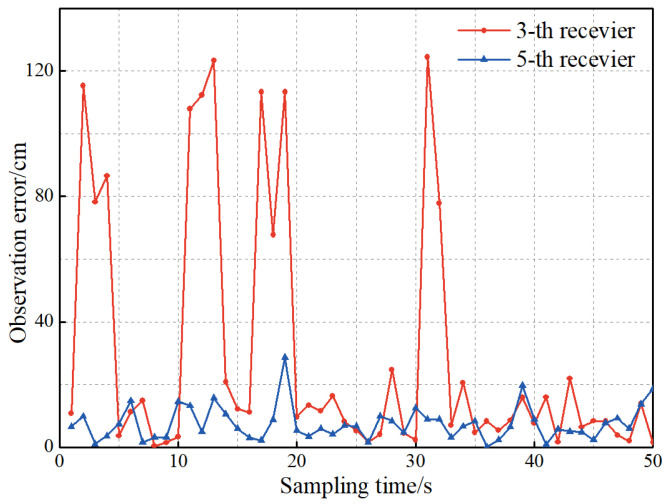
The observation error of the third and fifth receivers.

**Figure 11 sensors-23-02669-f011:**
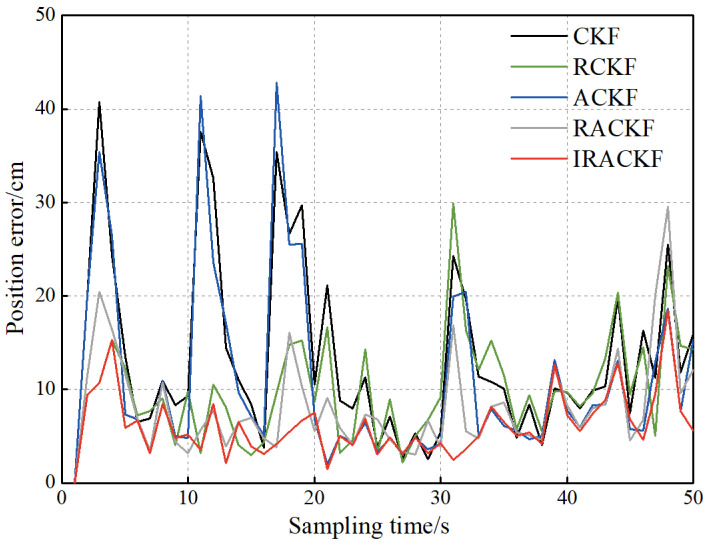
The position errors of filtering.

**Figure 12 sensors-23-02669-f012:**
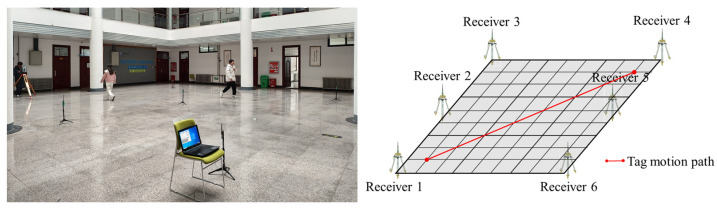
The experimental scene and tag movement trajectory.

**Figure 13 sensors-23-02669-f013:**
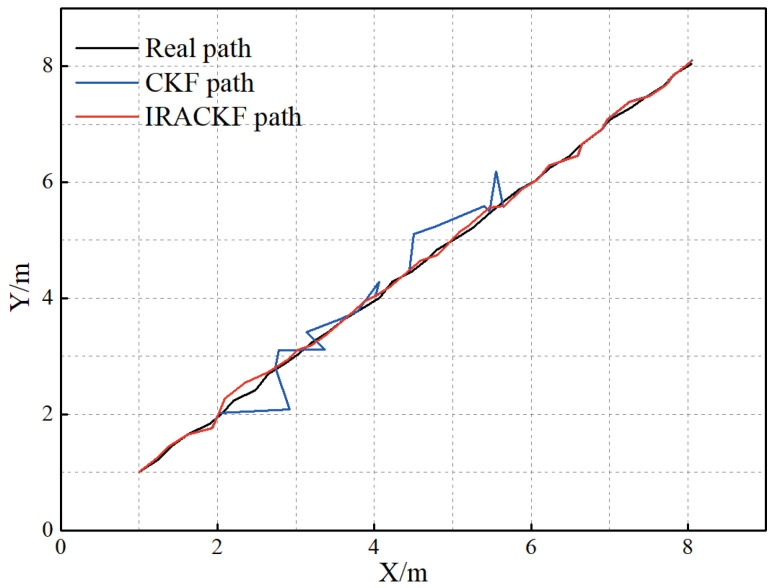
The tag movement trajectory for each filtering solution.

**Figure 14 sensors-23-02669-f014:**
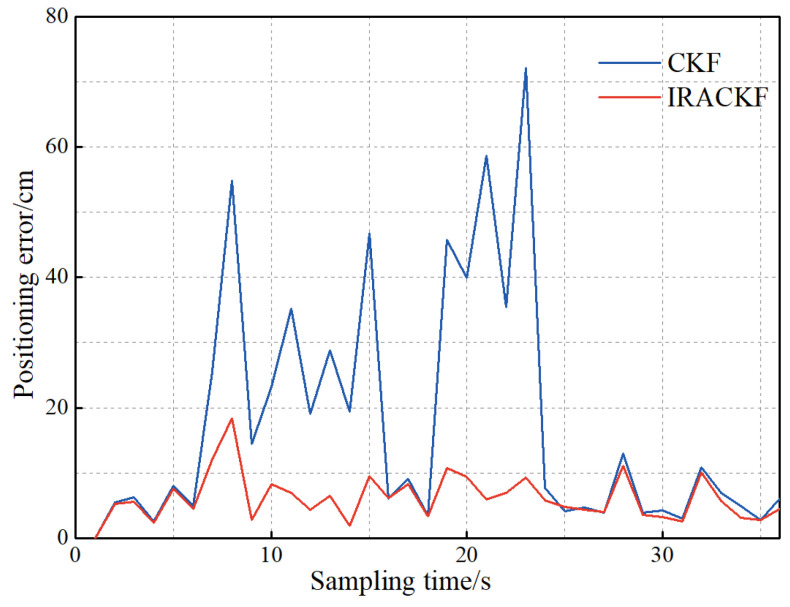
The position error of each filter.

## Data Availability

Not applicable.

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
