# Peer review of "UWB Localization Based on Improved Robust Adaptive Cubature Kalman Filter"

_sensors, 2023, doi:10.3390/s23052669_

Round 1
Reviewer 1 Report
The authors propose a new filter called "improved robust adaptive cubature Kalman filter (IRACKF)" useful to minimize localization errors in NLOS conditions. The idea of the authors is very innovative, but some adjustment must be made to the paper:
1. The authors should extend section 2, detailing the concepts related to the functioning of the TDOA in the NLOS case
2. There is the typos in the text. Authors should check them
3. In paragraph 4.1, when the authors describe the NLOS recognition algorithm based on the sliding window and the "polynomial fitting", the authors speak of a "threshold value". How is this “threshold value” defined?
4. Is the test area size, where the receivers are placed, significant? By varying the test area size, do you get the same performance?
5. Should the authors clarify the concept of sampling period/s indicated in the abscissa axis of figures 6-10?
Reviewer 2 Report
UWB technology is widely used to solve the localization problem in indoor environments.
However, it is well known that UWB technology is prone to non-line-of-sight (NLOS) problems.
In this article, the authors present an algorithm of a signal processing technology that improves the error in position determination.
The article is well written and well documented, even in the experimental part.
So, in general, I have no objection to its publication.
However, there is one point on which I would like to offer the authors a reflection.
The authors state that to simulate the used algorithm (IRACKF), they use Matlab 2019b on a computer equipped with a 3.4 GHz CPU and 8 GB RAM.
From a PC's point of view, the sampling rates you are using are so low that you can do everything in real-time.
But in the case of a actual implementation, the embedded system calculating the position will hardly be a desktop PC (maybe a SBC or, more likely, a microcontroller system, as in the case of the DWM1000 receiver you used to derive the probability density distribution of NLOS error).
In this case, the error reduction you obtained may not compensate for the higher computational cost of the proposed algorithm, both in terms of time and energy consumed.
Reviewer 3 Report
The authors presented IRACKF for UWB localization. The topic is interesting and widely recognized in localization research methods. I have the following comments for the authors:
1. The authors's contribution is not highlighted. robust adaptive cubature KF is already developed. therefore, the authors should clarify what improvement they are proposing in this paper, in the abstract and in the main section.
2. In (6), S=HPH'+R?
3. The authors should also discuss in detail the robust adaptive CKF in a new section so that the readers can compare the difference between the both. such as:
Robust Adaptive Cubature Kalman Filter and Its Application to Ultra-Tightly Coupled SINS/GPS Navigation System [2018].
4. what object was used as tag trajectory in the experiment? similarly, what are obstacles?
5. the title suggests the complex environment, but the experimental results doesn't show any complexity except the NLOS. Although, UWB is able to localize the tag in NLOS scenario well.
6. title should also be modified to "UWB localization based on improved robust adaptive cubature kalman filter"
7. The authors should also write the pseudo-code of the algorithm.
8. Overall, novelty is low. The authors should improve the novelty and contribution.
Thank you and best regards
Round 2
Reviewer 3 Report
The authors have addressed all comments and responded as per requirements. Therefore, I have no further comments. I would like to suggest an acceptance of this paper.
Thank you